# Stability of Pt-Adsorbed CO on Catalysts for Room Temperature-Oxidation of CO †

**Frédéric C. Meunier** [1,*] **, Taha Elgayyar** [1] **, Kassiogé Dembélé** [2] **and Helena Kaper** [3,*]

1 Univ Lyon, Université Claude Bernard Lyon 1, CNRS, IRCELYON, 2 Av. Albert Einstein, 69626 Villeurbanne, France; tahaelgayyar1@gmail.com

2 Department of Inorganic Chemistry, Fritz-Haber-Institut der Max-Planck-Gesellschaft, Faradayweg 4-6, 14195 Berlin, Germany; kdembele@fhi-berlin.mpg.de

3 Laboratoire de Synthèse et Fonctionnalisation des Céramiques, UMR 3080, CNRS/Saint-Gobain CREE, Saint-Gobain Research Provence, 550 Ave Alphonse Jauffret, 84300 Cavaillon, France

* Correspondence: fcm@ircelyon.univ-lyon1.fr (F.C.M.); helena.kaper@saint-gobain.com (H.K.)

† This paper reports original data first presented at the invited lecture of the MDPI *Catalysts* webinar of the 20 April 2022.

**Abstract:** A large signal of gas-phase CO overlapping with those of adsorbates is often present when investigating catalysts by operando diffuse reflectance FT-IR spectroscopy. Physically removing CO(g) from the IR cell may lead to a fast decay of adsorbate signals. Our work shows that carbonyls adsorbed on metallic Pt sites fully vanished in less than 10 min at 30 °C upon removing CO(g) when redox supports were used. In contrast, a broad band assigned to CO adsorbed on oxidized Pt sites was stable. It was concluded that physically removing CO(g) at room temperature during IR analyses will most likely lead to changes in the distribution of CO(ads) and a misrepresentation of the Pt site speciation, misguiding the development of efficient low-temperature CO oxidation catalysts. A tentative representation of the nature of the Pt phases present depending on the feed composition is also proposed.

**Keywords:** CO; platinum; FT-IR; in situ; operando

## 1. Introduction

Carbon monoxide is a harmful pollutant [1–3], especially indoors or inside vehicle cabins, and many efforts have been devoted to oxidize CO to $CO_2$ at low temperatures. Au and Pt-based catalysts are particularly active at room temperature, and promoted by the presence of moisture [4–6]. FT-IR spectroscopy analyses of CO adsorption can unravel the structure of supported metal [7–13] and nanoalloy [14–16] particles, and the diffuse reflectance mode (DRIFTS) is widely used to carry out in situ and *operando* studies [17]. DRIFTS usually leads to a noticeable signal of gas-phase CO, as the optical pathlength in the dead-volume of most DRIFTS cells is large—sometimes greater than 3 cm [18]. The two-branch rotovibrational spectrum of CO(g) spans over 2250–2050 cm$^{-1}$ and may overlap with adsorbate bands, and has actually been sometimes mistaken for those [19,20]. Ideally, CO(g) spectrum should be subtracted from in-situ and *operando* DRIFTS spectra to facilitate spectral analysis. Yet, it is surprising that this is not done in most cases.

The signal of CO(g) can be mathematically subtracted to solely reveal the signal of adsorbed species by using a reference spectrum collected under identical reaction conditions over a matrix that does not adsorb CO, e.g., KBr or SiC [18]. It is crucial to use identical conditions because the shape of the rotovibrational spectrum of CO depends on temperature, and DRIFTS cells will usually exhibit strong temperature gradients in the dead-volume comprised between the crucible holding the sample and the cell windows [21].

In many cases, CO was physically removed from the feed to reveal adsorbate bands [22–26]. This leads to less relevant conditions that may result in sample structural modifications, in

addition to rapid desorption of weakly adsorbed CO species. In fact, CO species adsorbed on isolated cationic atoms ($Pt_{iso}$) on $TiO_2$-based catalysts appear to be weakly bonded ($\Delta_{ads}H = -87$ kJ/mol) [27]. TPD in Ar of CO pre-adsorbed at $-140\,^{\circ}C$ shows that CO has already fully desorbed from $Pt_{iso}$ at $5\,^{\circ}C$. This stresses that many studies may have missed such sites if the IR analysis was made at, or above, room temperature after removing CO. In contrast, CO bonds more strongly to metallic $Pt^0$ clusters ($\Delta_{ads}H = -116$ to $-193$ kJ/mol) or oxidic $Pt_{ox}$ clusters ($\Delta_{ads}H = -172$ to $-210$ kJ/mol), and these species would typically be observed even after removing CO(g) [27].

However, the case of metallic Pt is actually more complex. The rate of CO desorption at room temperature from Pt(111) single crystals is minute, but non-zero (and coverage-dependent) [28], and that from 1.1 nm Pt nanoparticles supported on alumina is even lower [29]. Yet, strongly adsorbed CO may react with the oxygen from the lattice of redox supports. In this respect, CO adsorbed on $CeO_2$-supported metallic Pt nanoparticles was shown to be converted to $CO_2$ at room temperature by reacting with oxygen from ceria [30] or from PtOx species [6]. In the latter case, the carbonyl signal dropped by 90% within 60 s once gas-phase CO was removed from the feed [6]. Note that in both cases of these ceria-supported samples [6,30], no IR bands corresponding to CO adsorbed on isolated Pt cations or oxidized clusters could be observed, only metallic Pt-bond CO was present.

The present work aims at estimating: (i) the number of Pt-carbonyl species present at the surface of $CeO_2$- and $TiO_2$-based materials that are active for CO oxidation at room temperature; and (ii) the stability of the corresponding adsorbates around room temperature. The samples used here exhibits a more complex IR signal than those reported in previous studies [6,30], indicating the presence of various metallic and cationic Pt species. It must be stressed that the present study did not aim at determining the exact nature of the bands observed, particularly those located around 2090 $cm^{-1}$ on $Pt/CeO_2$, the assignment of which is still controversial [31]. The effect of the presence of molecular oxygen will also be assessed. These results will provide a rational basis for the relevance of carrying out IR analyses of CO adsorption over similar materials in the absence of CO(g).

## 2. Experimental Section

The titania-supported Pt sample was prepared using a wet impregnation method. $Pt(NO_3)_2$ (Heraeus) was dispersed in isopropanol. After addition of $TiO_2$ P25 (Evonik), the dispersion was sonicated for 15 min and then heated under stirring for 3 h. The solvent was then removed using rotational evaporation. The impregnated powder was calcined at $500\,^{\circ}C$ for 2 h.

The $Pt/CeO_2$ catalyst was prepared by a recently described procedure [32]. The sample was prepared in a one-step procedure resulting in Pt-doped ceria. Upon a reductive pretreatment under 40% $H_2$/He at $300\,^{\circ}C$ for 2 h, Pt is exsolved to the surface in the form of clusters.

The alumina-supported Pt sample was prepared using a wet impregnation method. The aqueous solution of the precursors of Pt (Platinic chloride hexahydrate, $H_2PtCl_6 \cdot 6H_2O$) from Sigma Aldrich was added to a suspension of 1 g of $\gamma$-alumina support (from Saint Gobain, $\gamma$-Alumina NORPRO 208 $m^2\,g^{-1}$) in 35 mL of distilled water, and stirred for 45 min at room temperature. A freshly prepared solution of $NaBH_4$ in 10 mL distilled water was added dropwise, and the stirring was maintained for 25 min. The $NaBH_4$/Pt ratio was 20. Finally, the suspension was filtered and washed several times with distilled water, and finally dried in an oven overnight at $90\,^{\circ}C$.

The TEM images of the Pt/alumina sample were obtained using a JEOL 2010 microscope with LaB6 source operated at 200 kV. The sample was suspended in EtOH solution and a drop of the sonicated suspension was deposited onto a carbon-coated Cu grid and solvent was evaporated. The TEM and HAADF-STEM image of $Pt/CeO_2$ and $Pt/TiO_2$ were prepared on lacey carbon films on copper grid, then analyzed on a JEOL ARM 200F operating at 200 kV. This microscope is equipped with a double spherical aberration correctors and GATAN Oneview and Orius cameras. In addition, a high-angle annular

dark-field (HAADF) detector was used, which maximized the collection of incoherent scattered electrons in scanning TEM (STEM) studies.

The high-purity gases He, 20% $O_2$/He and CO from Air Liquid were used. DRIFTS experiments were performed at 30 °C with a high temperature DRIFT cell (from Harrick) fitted with ZnSe windows using a Praying-Mantis assembly. A description and properties of the cell can be found in earlier references [33,34]. The spectrophotometer used was a Nicolet 8700 (ThermoFischer Scientific) fitted with a liquid-$N_2$ cooled MCT detector. The DRIFT spectra were recorded at a resolution of 4 cm$^{-1}$, and 32 scans were averaged. The DRIFTS spectra are reported as log (1/R), where R is the sample reflectance. This pseudo-absorbance gives a better linear representation of the band intensity against surface coverage than that given by the Kubelka-Munk function for strongly absorbing media such as those based on metals supported on oxides [35]. The contribution of gas-phase CO was subtracted using a CO(g) spectrum collected under the same experimental condition over KBr powder. The samples, previously reduced, were then exposed to 10% $O_2$/He (plus water if any) in the DRIFTS cell for 15 min before adding CO into the feed. This sequence leading to a mildly oxidized state of the sample is expected to represent that under which the catalyst would typically be operating for CO oxidation under ambient conditions.

## 3. Results and Discussion

The 1.3 wt.% Pt/alumina exhibited particle sizes mostly between 1 and 4 nm and a few larger ones (Figure 1). The particle size derived from the TEM analysis (surface-weighted diameter, measured over more than 200 particles) was 7.5 ± 1 nm, consistent with the size of the crystalline domains at 7.3 ± 1 nm determined by XRD (Scherrer broadening, not shown), corresponding to a Pt dispersion of about 15%.

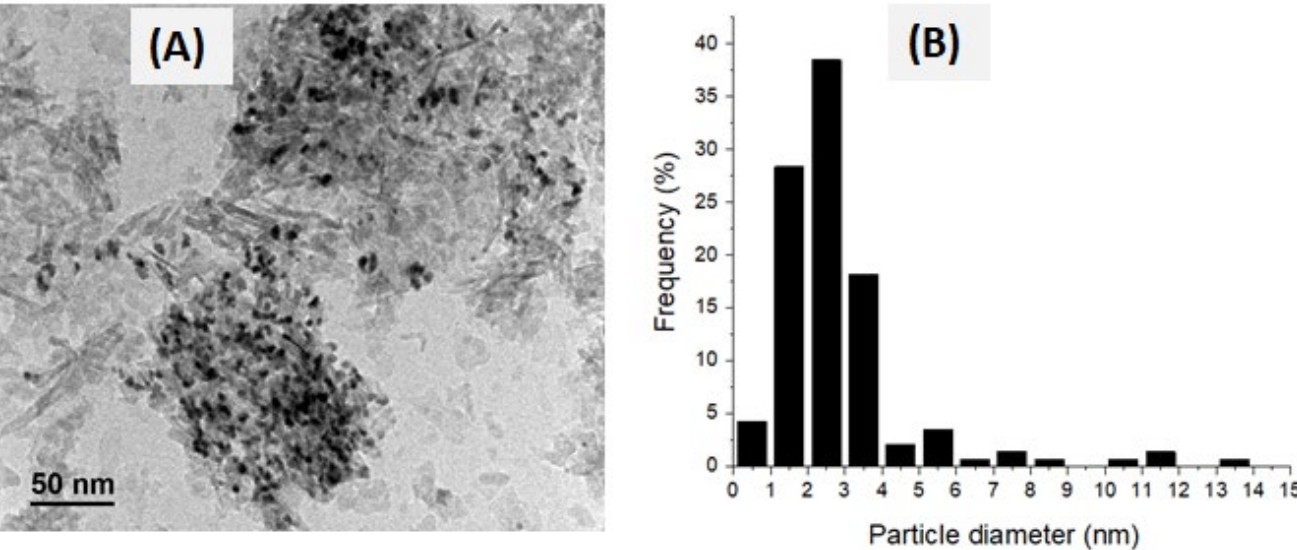

**Figure 1.** (**A**) TEM picture of the Pt/alumina and (**B**) corresponding particle size distribution.

The 1.3 wt% Pt/$TiO_2$ catalyst shows Pt nanoparticles well-distributed over the surface (Figure 2). The particle sizes range between 0.5 and 2.5 nm, with a mean size of 1.1 nm (measured on 231 particles).

For the 1.3 wt.% Pt/$CeO_2$ catalyst, Pt clusters can be found on the surface (Figure 3), but it was impossible to draw a precise particle size distribution, as part of the platinum was oxidized [6], and likely present as few-atoms clusters or rafts, leading to a poor contrast on supports such as ceria, with cerium having a high atomic number. It is assumed from previous studies realized over similar samples [32] that Pt dispersion was close to 100%. These particles partially agglomerate into nanoparticles (less than 2 nm in diameter) under reaction conditions (Figure 3C), similar to what has been observed in the literature on similar samples [36].

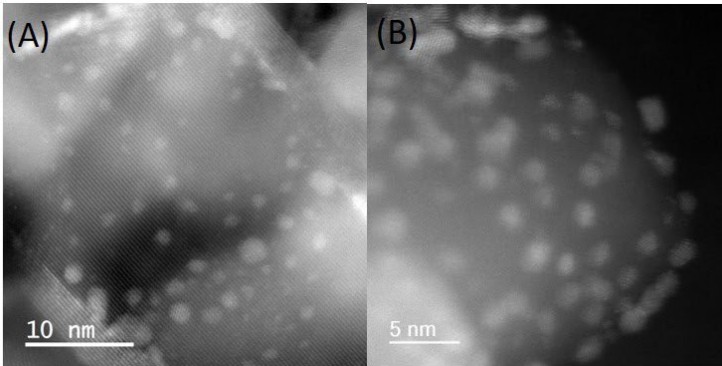
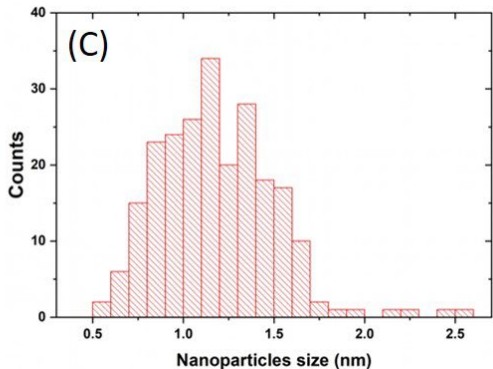

**Figure 2.** (**A**,**B**) HAADF-STEM images of the 1.3 wt.% $Pt/TiO_2$ and (**C**) corresponding particle size distribution.

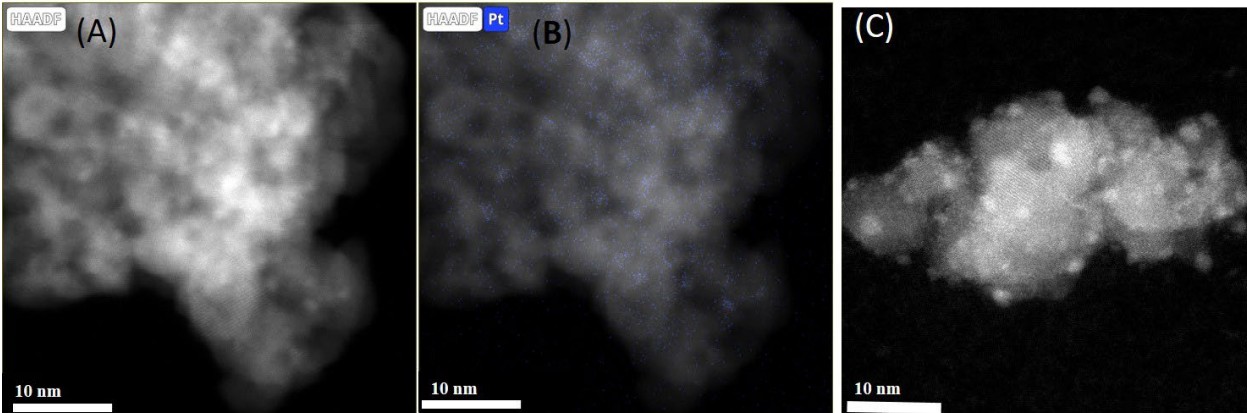

**Figure 3.** (**A**) HAADF-STEM images of the 1.3 wt.% $Pt/CeO_2$ and (**B**) STEM-EDX map analysis on the same area indicating the Pt distribution. (**C**) Pt-particles on 1.3 wt.% $Pt/CeO_2$ after reaction.

The catalytic activity measured over the Pt/alumina at steady-state under the $CO + O_2$ feed at 30 °C was negligible, in accordance with the fact that CO poison alumina-supported Pt at low temperatures [37]. The corresponding steady-state DRIFTS spectrum is shown in Figure 4 (red spectra). The main bands at 2085 $cm^{-1}$ was asymmetric, suggesting the presence of at least two different linear Pt species. The band at 1845 $cm^{-1}$ can be unambiguously assigned to bridged CO on metallic Pt sites [8,10]. Therefore, part of the asymmetric 2085 $cm^{-1}$ band can be assigned to linear CO adsorbed on metallic Pt nanoparticles.

The linear CO band slightly shifted (Figure 4A), and its intensity remained constant (Figure 5A) under He, following the removal of both $O_2$ and CO at 30 °C. The shift may have been due to a re-ordering of adsorbed CO at the surface of the nanoparticles, as about half the bridged CO species were gradually removed (Figure 4A).

The carbonyl band evolution was quite different when $O_2$ was left in the feed following CO removal. The intensity of the band of linear CO rapidly dropped by half (Figures 4B and 5A), and bridged CO fully vanished within 5 min (Figure 4B); this suggests that these carbonyls were displaced or oxidized by $O_2$. Interestingly, a stable broad linear CO band was then observed around 2085 $cm^{-1}$. This band may be assigned to CO adsorbed on partially oxidized Pt. Several authors have proposed that CO could bound strongly to PtOx species and be unreactive [9,27,38], although the exact nature of the sites, e.g., single Pt cation or PtOx clusters, is still debated, as discussed in a recent review [31]. The main linear CO band on the Pt/alumina sample can thus be assigned to two different Pt species, one oxidized (2085 $cm^{-1}$) and one reduced (ca. 2065 $cm^{-1}$), with only the latter showing some reactivity in the presence of $O_2$ at 30 °C.

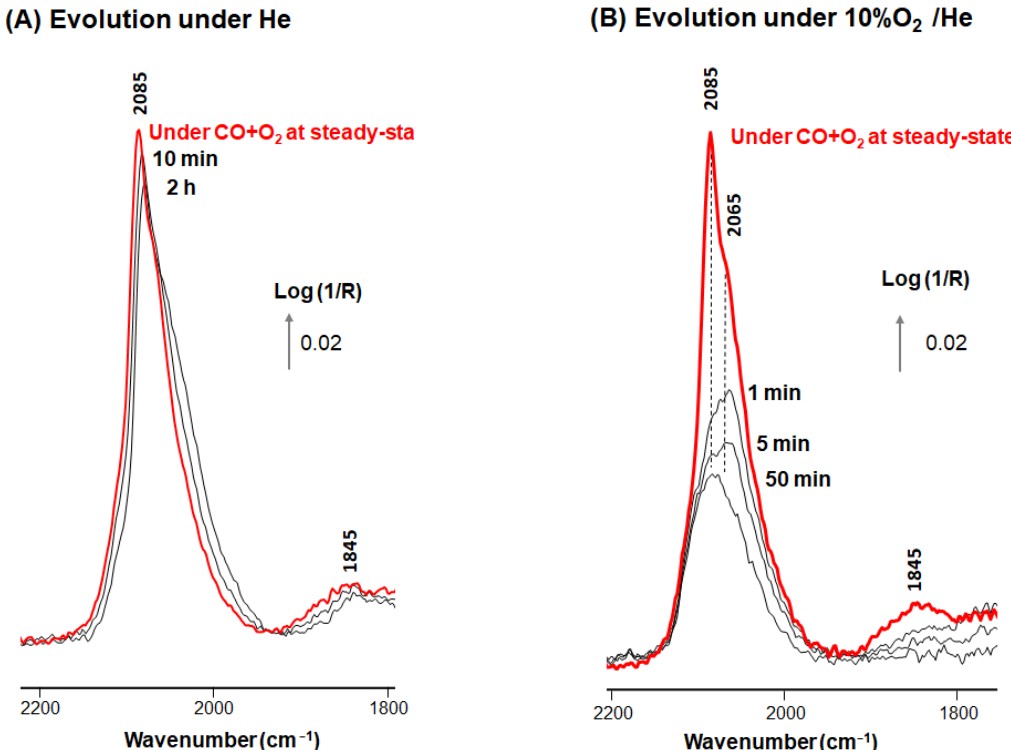

**Figure 4.** Evolution of DRIFTS spectra over $Pt/Al_2O_3$, recorded at 30 °C as a function of time under (**A**) He and (**B**) 10% $O_2$/He. Total flow rate: 100 mL/min. The sample was previously left 15 min under 10% $O_2$/He and then 15 min under 0.5% CO + 10% $O_2$/He to reach steady-state (Red spectrum).

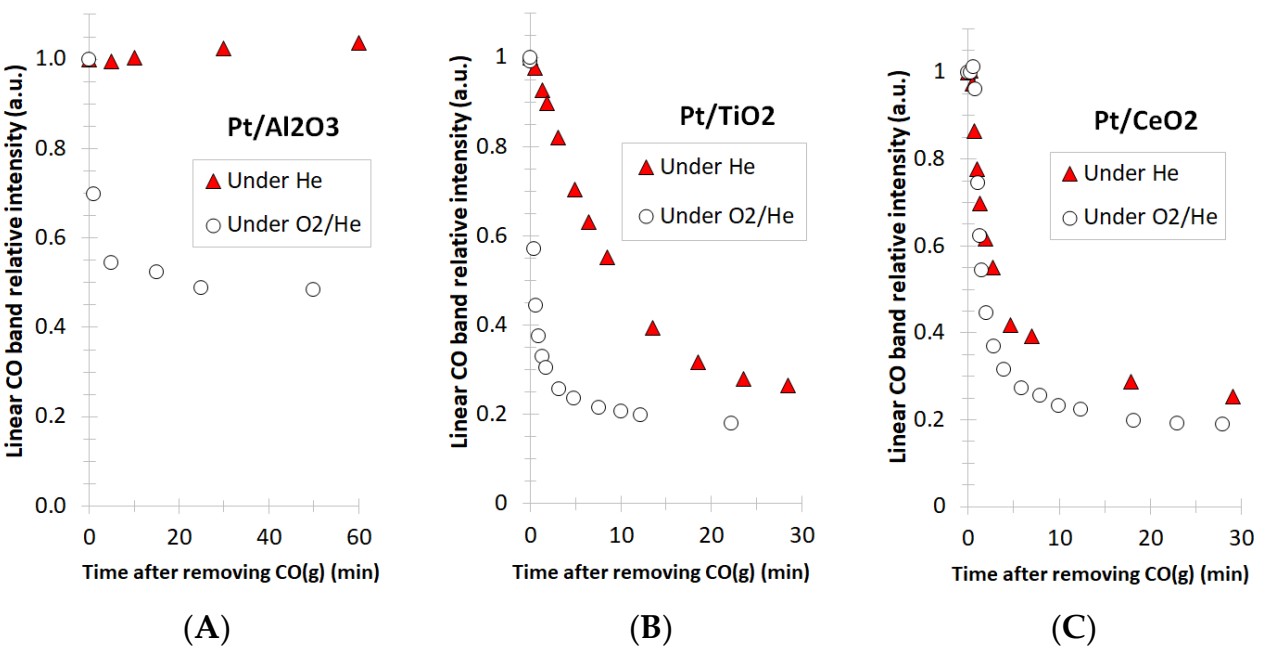

**Figure 5.** Relative area of the linear carbonyl DRIFTS band as a function time following the removal of CO from the feed measured over Pt supported on (**A**) alumina, (**B**) titania and (**C**) ceria. The sample was previously left 15 min under 10% $O_2$/He, and then 15 min under 0.5 % CO + 10% $O_2$/He. Feed: 100 mL min$^{-1}$ of (▲) He and (○) 10% $O_2$/He.

The Pt/TiO$_2$ sample presented a measurable catalytic activity at 30 °C, unlike the alumina-based sample, with a TOF of about $4.0 \times 10^{-3}$ s$^{-1}$. This TOF is similar to the highest TOFs reported by Hong and co-workers (i.e., $5.5 \times 10^{-3}$ s$^{-1}$) at 25 °C on Pt/TiO$_2$ catalysts [39]. The steady-state DRIFTS spectrum measured over the Pt/TiO$_2$ under the CO + O$_2$ feed at 30 °C appeared to be more complex than that observed on the Pt/alumina. At least three main bands of linear CO could be observed at 2112, 2088 and 2072 cm$^{-1}$ (Figure 6). Bridged carbonyls were also present at 1839 cm$^{-1}$.

**(A) Evolution under CO + 10% O$_2$ /He**

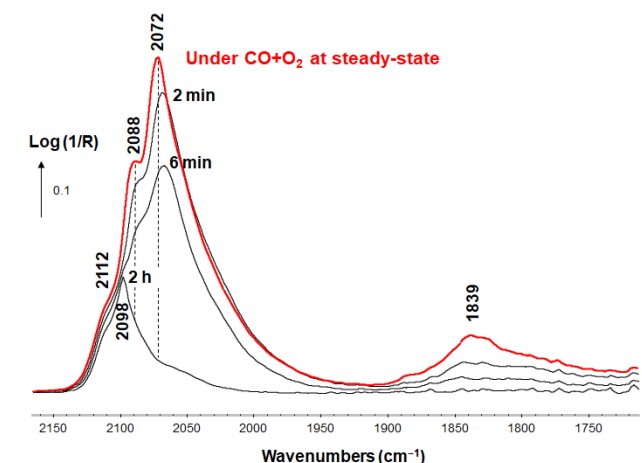

**(B) Evolution under He**

**(C) Evolution under 10% O$_2$ /He**

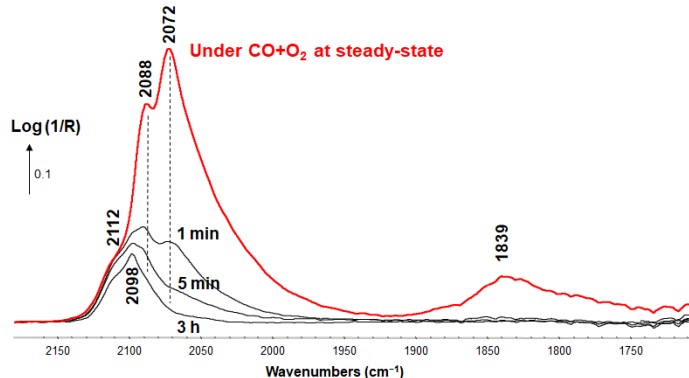

**Figure 6.** Evolution of DRIFTS spectra over Pt/TiO$_2$ recorded at 30 °C as a function of time under (**A**) 0.5% CO + 10% O$_2$/He for 15 min. The sample was then exposed to either (**B**) pure He or (**C**) 10% O$_2$/He. Total flow rate: 100 mL/min. The sample was initially left 15 min under 10% O$_2$/He.

The evolution of the DRIFTS spectra at 30 °C over the Pt/TiO$_2$ under 10% O$_2$/He sample following the introduction of CO was used to unravel the nature of the bands present (Figure 6A). The bands at 2112 and 2088 cm$^{-1}$ were first observed, with only very few bridged CO, which are typical of metallic Pt. The bands at 2112 cm$^{-1}$ is assigned to CO adsorbed on PtOx species, as proposed earlier [8,27,38]. The bands at 2088 cm$^{-1}$ can be assigned to CO adsorbed on metallic Pt partly covered with oxygen atoms [40,41]. Larger bands 2072 and 1839 cm$^{-1}$ were subsequently observed, and are assigned to linear and bridged CO adsorbed on metallic Pt nanoparticles, respectively.

The fact that the two bands related to metallic Pt (2072 and 1839 cm$^{-1}$) only gradually developed suggests that the corresponding sites were not initially available for adsorption, most likely because the Pt nanoparticles were initially oxidized to PtO$_2$. A similar gradual build-up of linear CO on metallic Pt was observed on Pt/CeO$_2$, for which the initial presence of PtO$_2$ was observed by NAP-XPS [6]. Pt$^{4+}$ species on TiO$_2$ were shown not to adsorb CO, as those are coordinatively saturated [42]. These data show that the speciation of PtOx species on titania is complex, with some species being reduced by CO at room temperature, while others cannot be. These assignments are gathered in Table 1.

**Table 1.** Tentative IR band assignments related to CO adsorption. "L" refers to linearly adsorbed CO and "B" to bridged CO between two Pt atoms.

| Wavenumber (cm$^{-1}$) | | References |
|---|---|---|
| - | PtO$_2$: does not adsorb CO at RT | [42] |
| 2112–2098 | L-CO on PtOx | [8,9,27,38] |
| 2088 | L-CO on O-covered metallic Pt | [40,41] |
| 2075–2060 | L-CO on Pt$^0$ | [8,10] |
| 1850–1830 | B-CO on Pt$^0$ | [8,10] |

The carbonyl bands at 2088, 2072 and 1839 cm$^{-1}$ on Pt/TiO$_2$ all decayed rapidly when both CO and O$_2$ were removed from the He feed, in contrast to the case of the alumina-supported Pt (Figures 5B and 6B). The decay was even faster when only CO was removed the feed, keeping 10% O$_2$ in the He stream (Figures 5B and 6C). These data indicate that CO adsorbed on metallic Pt was readily displaced or oxidized by gas-phase O$_2$, as already reported [29], and possibly oxidized by O atoms from the titania lattice. More work would be needed to ascertain this latter point, which is outside the scope of this contribution.

In contrast, a group of bands with maxima around 2112 and 2098 cm$^{-1}$ were stable for over 3 h both under He and 10% O$_2$/He (Figures 5B and 6B,C). The band at 2098 cm$^{-1}$ is assigned to CO adsorbed on a PtOx species, similarly to that at 2112 cm$^{-1}$. These CO species are thus strongly bound to unreactive oxidized Pt sites at 30 °C. The same conclusion was proposed by Hong and co-workers [39], who proposed that metallic Pt was the only active site at room temperature, and that the TOF value increased with the fraction of metallic Pt.

The Pt/CeO$_2$ sample was active for CO oxidation and the TOF measured at 30 °C after 15 min was $5.8 \times 10^{-3}$ s$^{-1}$. This value is about a third of that reported on some of the best Pt/CeO$_2$ catalysts used for CO oxidation at room temperature exhibiting TOFs around $2 \times 10^{-2}$ s$^{-1}$ [6,30]. The sample prepared by us in reference [25] was actually prepared by an impregnation method, and contained more nanoparticles, while the present sample contains smaller clusters, in addition to Pt doping the ceria lattice.

The corresponding steady-state DRIFTS spectrum is shown in Figure 7 (red spectra). The spectrum presented two linear CO bands at 2098 (shoulder) and 2074 cm$^{-1}$ and a bridge CO at 1831 cm$^{-1}$. The band at 2098 cm$^{-1}$ band can be assigned to linear CO adsorbed on PtOx species, while those at 2074 and 1831 cm$^{-1}$ can be assigned to CO on metallic Pt (Table 1). It is interesting to note that for Pt/CeO$_2$, the signal decay following CO removal

was fast, and essentially identical whether $O_2$ was present or not (Figures 5C and 7). Only the metallic Pt-bound CO were removed, while PtOx-bound CO was stable, as for the case of Pt/TiO$_2$.

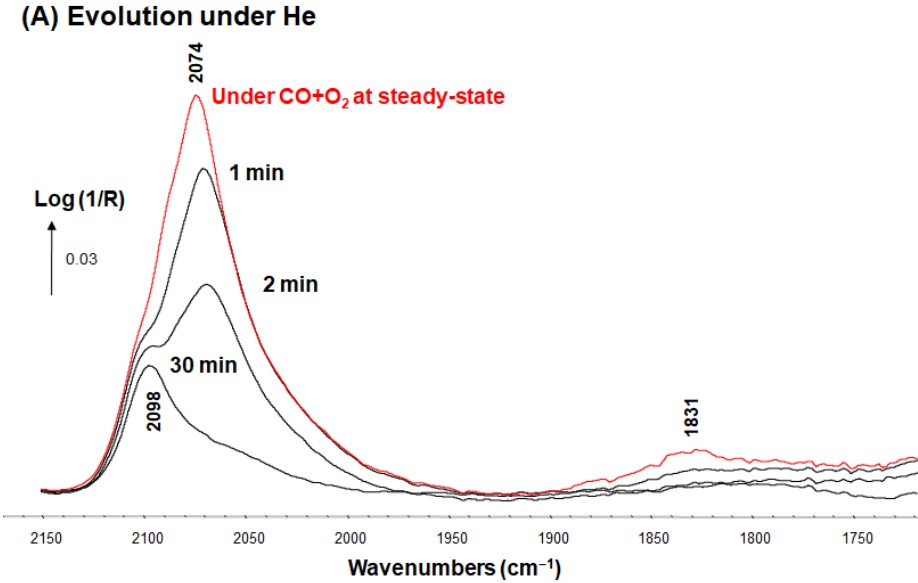

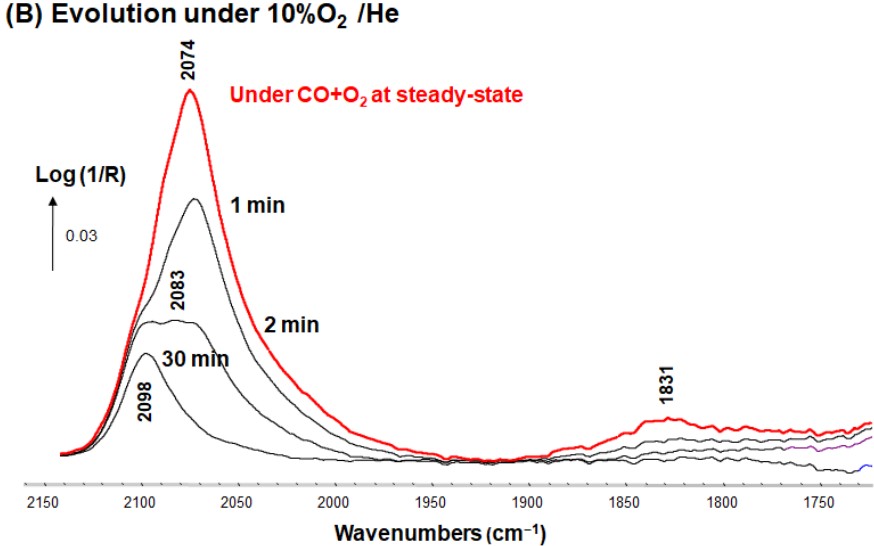

**Figure 7.** Evolution of DRIFTS spectra over Pt/CeO$_2$ recorded at 30 °C as a function of time under (**A**) pure He and (**B**) 10% O$_2$/He. Total flow rate: 100 mL/min. The sample was previously left 15 min under 10% O$_2$/He, and then 15 min under 0.5% CO + 10% O$_2$/He to reach steady-state (Red spectra).

The metallic Pt-bound CO was quantitatively shown to be the main reaction intermediate in CO oxidation on Pt/CeO$_2$, which could react with oxygen atoms from ceria lattice [30] or PtOx species [6]. The fact that signal decay was identical whether or not O$_2$ was present indicates that the reacting O was primarily coming from ceria or PtOx (e.g., PtO$_2$) species, and not from gas-phase O$_2$.

These data stress that various Pt species are present under reaction conditions pertaining to CO oxidation at room temperature. One Pt nanostructure will be oxidized as PtO$_2$ in the presence of O$_2$ and no CO, while it will be reduced to metallic Pt when exposed to CO + O$_2$. The other Pt nanostructure always remains oxidized and strongly adsorbs CO. This PtOx-CO adduct is highly stable, and shows no reactivity or decomposition at room temperature.

These results agree with the finding of Datye and co-workers for CO oxidation on Pt/CeO$_2$ [38] that CO adsorbed on cationic Pt species were orders of magnitude less reactive than those present on metallic Pt nanoparticles. Similarly, Hong and co-workers showed that metallic Pt was an important part of the catalytic ensemble [39]. We recently showed that the presence of both metallic and oxidized Pt was actually beneficial to room temperature CO oxidation activity [6]. The simultaneous presence of metallic Pt and a redox phase (CeO$_2$ or PtO$_2$) is thus required to reach a significant steady-state CO oxidation activity at room temperature. These findings are schematically summarized in Figure 8.

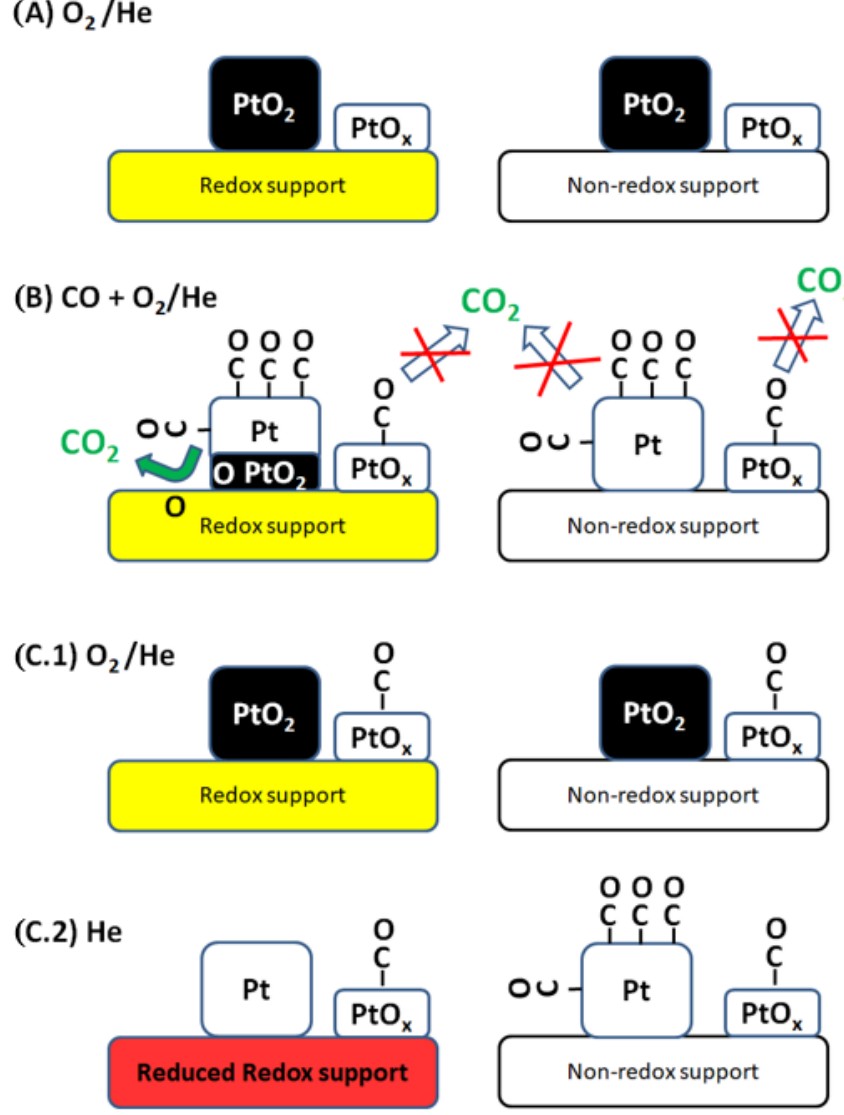

**Figure 8.** Schematic representation of the steady-state of various Pt nanostructures depending on the support and gases they are exposed to at room temperature. One of the Pt nanostructures, such as a large nanoparticle, is able to be fully oxidized or reduced depending on feed composition, while the other nanostructure, such as a cluster PtOx or an isolated Pt cation, remains oxidized. (**A**) All Pt phases are oxidized under O$_2$, (**B**) Only redox-based supports lead to a significant CO oxidation activity at room temperature. CO adsorbed on reduced Pt site reacts with oxygen from the redox support or from an adjacent oxidized Pt phase. (**C.1**) If only CO is removed from the feed, large Pt nanoparticle are reoxidised whatever the support, while CO remains strongly adsorbed on inert PtOx domains. (**C.2**) If both CO and O$_2$ are removed from the feed, the adsorbed CO left over large nanoparticles will react and leave the large Pt nanoparticles and redox support partly reduced. In the case of a non-redox support, all Pt linear sites remain covered with CO.

## 4. Conclusions

Physically removing gas-phase CO from the IR cell led to a fast decay of carbonyl signals, even on metallic Pt sites over redox supports. CO adsorbed on metallic Pt sites fully vanished in less than 10 min at 30 °C upon removing CO(g) when $CeO_2$ was used as a support. The rate of decay was hardly affected by the presence of $O_2$, suggesting that CO was converted to $CO_2$ using oxygen species from the support and/or $PtO_2$. In contrast, the presence of $O_2$ significantly accelerated $Pt^0$-bound carbonyl removal in the case of $Pt/TiO_2$. In the case of Pt/alumina, carbonyl signal decay was also observed in the presence of molecular oxygen, but not under helium. A broad band assigned to CO adsorbed on oxidized Pt sites, probably PtOx clusters, was stable in both cases. It is concluded that physically removing CO(g) at room temperature during IR analyses leads to a misrepresentation of the Pt site speciation.

**Author Contributions:** F.C.M. and H.K. conceptualized the work. H.K. prepared the ceria- and titania-based catalysts. F.C.M. and T.E. carried out the IR analyses. K.D. carried out the TEM analyses. F.C.M. obtained the funding and prepared the draft of the paper, and all authors contributed to the final version. All authors have read and agreed to the published version of the manuscript.

**Funding:** Part of this research was funding by the ANR (DECOMPNOx project, ANR-18-CE07-0002-01).

**Data Availability Statement:** Not applicable.

**Acknowledgments:** Ranin Atwi is acknowledged for the preparation of the alumina-supported sample.

**Conflicts of Interest:** The authors declare no competing financial interest.

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
