# Peer review of "Stability of Pt-Adsorbed CO on Catalysts for Room Temperature-Oxidation of CO†"

_catalysts, doi:10.3390/catal12050532_

Round 1

Reviewer 1 Report

In this article, the catalysts were successfully synthesized for room temperature-oxidation of CO. And, it was found that physically removing CO(g) at room temperature during IR analyses will most likely lead to changes in the distribution of CO(ads) and a misrepresentation of the Pt site speciation, misguiding the development of efficient low-temperature CO oxidation. This has certain significance. However, there are still some matters. After careful evaluation of the manuscript, detailed comments are as follows:

  1. There are many ways to detect CO. Please introduce. Besides, please give a brief introduction to CO, such as its harm, etc. Some new references should be added. Such as Molecular Catalysis, 2021, 509, 111633; Journal of Catalysis 409 (2022) 1–11
  2. In this paper, the tenses of statements often change, resulting in errors, and the unit symbol is also wrong in some places. Please check the correction.
  3. In this paper, there is no correlation diagram such as XRD to prove the successful synthesis of the material. Please supplement XRD to further prove the successful synthesis of the sample
  4. The corresponding particle size distribution of Pt/CeO2 is impossible to draw in this paper. Please explain the reasons.
  5. In Figure 4, “The main linear CO band on the Pt/alumina sample can thus be assigned to two different Pt species, one oxidized (2085 cm-1) and one reduced (ca. 2065 cm-1)”, it is only the author's reasonable speculation. Please take other methods for further verification.
  6. In evolution of DRIFTS, the three catalysts didn’t expose in the completely same conditions. Please explain.
  7. Specific surface area is important for materials. Please give N2 adsorption isotherm and pore size distribution curve. The isotherm and hysteresis loop need be explained. BJH adsorption or desorption are selected for mesoporous ones. Please refer to and cite the literature for drawings and explanations. Molecular Catalysis, 522 (2022) 112226

Author Response

In this article, the catalysts were successfully synthesized for room temperature-oxidation of CO. And, it was found that physically removing CO(g) at room temperature during IR analyses will most likely lead to changes in the distribution of CO(ads) and a misrepresentation of the Pt site speciation, misguiding the development of efficient low-temperature CO oxidation. This has certain significance. However, there are still some matters. After careful evaluation of the manuscript, detailed comments are as follows:

1-There are many ways to detect CO. Please introduce. Besides, please give a brief introduction to CO, such as its harm, etc. Some new references should be added. Such as Molecular Catalysis, 2021, 509, 111633; Journal of Catalysis 409 (2022) 1–11

Authors’ reply: The reviewer is correct and we have added the suggested references, plus two others, to which the reader is referred for more information on the harmfulness of CO. The following sentences were added at the start of the introduction: “Carbon monoxide is a harmful pollutant [1,2,3], especially indoor or inside vehicle cabins, and many efforts have been devoted to oxidise CO to CO2 at low temperatures. Au and Pt-based catalysts are particularly active at room temperature and promoted by the presence of moisture [3,4,5]”

New references:

1 Bi, F.; Zhang, X.; Du, Q.; Yue, K.; Wang, R.; Li, F.; Liu, N.; Huang, Y. Influence of pretreatment conditions on low-temperature CO oxidation over Pd supported UiO-66 catalysts. Molecular Catalysis, 2021, 509, 111633.

2 Di, M.; Simmance, K.; Schaefer, A.; Feng, Y.; Hemmingsson, F.; Skoglundh, M.; Bell, T.; Thompsett, D.;  Jensen, L.I.A.; Blomberg, S.; Carlsson, P.-A. Chasing PtOx species in ceria supported platinum during CO oxidation extinction with correlative operando spectroscopic techniques, J. Catalysis, 2022, 409, 1-11.

3 Rangel, R.; González-A, E.; Solís-García, A.; Zepeda, T.A.; Galván, D.H.; Gómez-Cortés, A.; Díaz, G. Pt and Ir supported on mixed Ce0.97Ru0.03O2 oxide as low-temperature CO oxidation catalysts. Catal. Today, 2022, 392–393, 3-12.

4 Kale, M.J.; Gidcumb, D.; Gulian, F.J.; Miller, S.P.; Clark, C.H.; Christopher, P. Evaluation of platinum catalysts for naval submarine pollution control. Appl. Catal. B: Env. 2017, 203, 533-540.

5 Saavedra J.; Doan, H.A.; Pursell, C.J.; Grabow, L.C.; Chandler, B.D. The critical role of water at the gold-titania interface in catalytic CO oxidation. Science 2014, 345, 1599-1602.

2- In this paper, the tenses of statements often change, resulting in errors, and the unit symbol is also wrong in some places. Please check the correction.

Authors’ reply: The manuscript was further checked, thank you for raising this issue.

3-In this paper, there is no correlation diagram such as XRD to prove the successful synthesis of the material. Please supplement XRD to further prove the successful synthesis of the sample

Authors’ reply: We are not sure to understand this point. Most of the Pt nanoparticles here are smaller than 3 nm so we do not believe that XRD analyses would provide more relevant information than the TEM data already reported.

4- The corresponding particle size distribution of Pt/CeO2 is impossible to draw in this paper. Please explain the reasons.

Authors’ reply: as shown in earlier papers (ref 27), the Pt dispersion is close to 100 % and a significant proportion of Pt is oxidized, possibly under the form of few-atom clusters and rafts. This leads poor contrast during TEM analysis on support such as ceria and make impossible obtained a clear particle size distribution. The following sentence has been added to the text: “, because of part of the platinum was oxidized [ref 5], likely present as few-atoms clusters or rafts, leading to a poor contrast on supports such as ceria, cerium having a high atomic number.”.

5- In Figure 4, “The main linear CO band on the Pt/alumina sample can thus be assigned to two different Pt species, one oxidized (2085 cm-1) and one reduced (ca. 2065 cm-1)”, it is only the author's reasonable speculation. Please take other methods for further verification.

Authors’ reply: the fact that the first band at 2065/cm disappear in parallel to that at 1845/cm (bridged Pt2-CO) is a strong argument that the 2065/cm is due to Pt on metallic Pt. The tentative assignment of the 2085/cm band is indeed less certain, as is already stated in the paper, and is well in lign with the references given (8,26,37)

6- In evolution of DRIFTS, the three catalysts didn’t expose in the completely same conditions. Please explain.

Authors’ reply: we gave additional DRIFTS spectra in the case of the Pt/TiO2 to detail the growth of the complex IR signal that can be obtained over this type of redox support. It was not necessary on the case of Pt/alumina that gives simpler spectra. We did not repeat the detail in the case of Pt/CeO2, because it was similar to that observed on Pt/TiO2 and did not bring additional insights.

7- Specific surface area is important for materials. Please give N2 adsorption isotherm and pore size distribution curve. The isotherm and hysteresis loop need be explained. BJH adsorption or desorption are selected for mesoporous ones. Please refer to and cite the literature for drawings and explanations. Molecular Catalysis, 522 (2022) 112226

Authors’ reply: pore distribution is indeed important when kinetic data are considered, but in the present case only the nature of the adsorption of CO on Pt is discussed. We therefore feel that the pore structure of the support is not relevant to the present study.

Reviewer 2 Report

This manuscript represents a sort of methodological warning during CO adsorption studies on metallic sites like Pt supported over oxides.

The work is really interesting (and unusual in a way) and underline how the interpretation of FT-IR or DRIFT spectra after probe molecules interactions is sometimes complicate by the experimental methodology.

The main question is: how the authors suggest they can overcome the problem they themselves highlight? In other world, is it possible to use the CO adsorption DRIFT without blinding or only using an operando FT-IR (i.e. with gas flowing) we can reach a correct interpretation?

Author Response

This manuscript represents a sort of methodological warning during CO adsorption studies on metallic sites like Pt supported over oxides.

The work is really interesting (and unusual in a way) and underline how the interpretation of FT-IR or DRIFT spectra after probe molecules interactions is sometimes complicate by the experimental methodology. The main question is: how the authors suggest they can overcome the problem they themselves highlight? In other world, is it possible to use the CO adsorption DRIFT without blinding or only using an operando FT-IR (i.e. with gas flowing) we can reach a correct interpretation?

Authors’ reply: thank you for the kind words. The monitoring of CO adsorption by IR is clearly a valuable tool, especially for catalysts that are meant to convert CO. The fact that adsorbates can react at room temperature cannot be avoided and it is part indeed of the objective that seeks high activity catalysts for CO removal at room temperature. We do not see any alternative to monitoring the structure of the surface sites in the presence of CO.

Reviewer 3 Report

The manuscript report interesting information on the subject. In my opinion only minor revision are required:

The format of the references it is not the Catalysts style.

These relevant papers should be cited in the introduction (Appl. Catal. A 520 (2016) 82–91; Catalysis Today 392-393 (2022) 3-12; Journal of Electroanalytic Chemistry 886 (2021) 115149.

What is the influence of the surface area and the pore size distrubution in this study?

Author Response

The manuscript report interesting information on the subject. In my opinion only minor revision are required:

1- The format of the references it is not the Catalysts style.

Authors’ reply: thank you for highlighting this. We shall change the reference formatting if this is necessary.

2- These relevant papers should be cited in the introduction (Appl. Catal. A 520 (2016) 82–91; Catalysis Today 392-393 (2022) 3-12; Journal of Electroanalytic Chemistry 886 (2021) 115149.

Authors’ reply: we have added the second reference that is most relevant to this study. Rangel, R.; González-A, E.; Solís-García, A.; Zepeda, T.A.; Galván, D.H.; Gómez-Cortés, A.; Díaz, G. Pt and Ir supported on mixed Ce0.97Ru0.03O2 oxide as low-temperature CO oxidation catalysts. Catal. Today, 2022, 392–393, 3-12. We actually noted some IR data misinterpretation in this paper, that we shall note in Pubpeer and inform the authors (bands of gas-phase CO at 2167 + 2124 /cm were wrongly assigned to CO adsorbed on Ce3+ and Ce4+)

3- What is the influence of the surface area and the pore size distrubution in this study?

Authors’ reply: as indicated to reviewer 1, pore distribution is indeed important when kinetic data are considered, but in the present case only the nature of the adsorption of CO on Pt is discussed. We therefore feel that the pore structure of the support is not relevant to the present study. The surface area of the support would affect the dispersion of the metal, but again this is not the point of the present study.

Round 2

Reviewer 1 Report

accepted